# On the computational complexity of curing non-stoquastic Hamiltonians

Milad Marvian[1,2,3], Daniel A. Lidar[2,3,4,5] & Itay Hen [3,4,6]

Quantum many-body systems whose Hamiltonians are non-stoquastic, i.e., have positive off-diagonal matrix elements in a given basis, are known to pose severe limitations on the efficiency of Quantum Monte Carlo algorithms designed to simulate them, due to the infamous sign problem. We study the computational complexity associated with 'curing' non-stoquastic Hamiltonians, i.e., transforming them into sign-problem-free ones. We prove that if such transformations are limited to single-qubit Clifford group elements or general single-qubit orthogonal matrices, finding the curing transformation is NP-complete. We discuss the implications of this result.

[1] Research Laboratory of Electronics, Massachusetts Institute of Technology, Cambridge, MA 02139, USA. [2] Department of Electrical and Computer Engineering, University of Southern California, Los Angeles, CA 90089, USA. [3] Center for Quantum Information Science & Technology, University of Southern California, Los Angeles, CA 90089, USA. [4] Department of Physics and Astronomy, University of Southern California, Los Angeles, CA 90089, USA. [5] Department of Chemistry, University of Southern California, Los Angeles, CA 90089, USA. [6] Information Sciences Institute, University of Southern California, Marina del Rey, CA 90292, USA. Correspondence and requests for materials should be addressed to M.M. (email: mmarvian@mit.edu)

The "negative sign problem", or simply the "sign problem"[1], is a central unresolved challenge in quantum many-body simulations, preventing physicists, chemists, and material scientists alike from being able to efficiently simulate many of the most profound macroscopic quantum physical phenomena of nature, in areas as diverse as high-temperature superconductivity and material design through neutron stars to lattice quantum chromodynamics. More specifically, the sign problem slows down quantum Monte Carlo (QMC) algorithms[2,3], which are in many cases the only practical method available for studying large quantum many-body systems, to the point where they become practically useless. QMC algorithms evaluate thermal averages of physical observables by the (importance-) sampling of quantum configuration space via the decomposition of the partition function into a sum of easily computable terms, or weights, which are in turn interpreted as probabilities in a Markovian process. Whenever this decomposition contains negative terms, QMC methods tend to converge exponentially slowly. Most disheartingly, it is typically the systems with the richest quantum`-mechanical behavior that exhibit the most severe sign problem.

In defining the scope under which QMC methods are sign-problem free, the concept of "stoquasticity", first introduced by Bravyi et al.[4], has recently become central. The most widely used definition of a local stoquastic Hamiltonian is

**Definition 1**[5] A *local Hamiltonian*, $H = \sum_{a=1}^M H_a$ is called stoquastic with respect to a basis $\mathcal{B}$, iff all the local terms $H_a$ have only non-positive off-diagonal matrix elements in the basis $\mathcal{B}$.

In the basis $\mathcal{B}$, the partition function decomposition of stoquastic Hamiltonians leads to a sum of strictly nonnegative weights and such Hamiltonians hence do not suffer from the sign problem. For example, in the path-integral formulation of QMC with respect to a basis $\mathcal{B} = \{b\}$, the partition function $Z$ at an inverse temperature $\beta$ is reduced to an $L$-fold product of sums over complete sets of basis states, $\{b_1\}, \ldots, \{b_L\}$, which are weighted by the size of the imaginary-time slice $\Delta\tau = \beta/L$ and the matrix elements of $e^{-\Delta\tau H}$. Namely, $Z = \prod_{l=1}^L \sum_{b_l} \langle b_l | e^{-\Delta\tau H} | b_{l+1} \rangle$, where $L$ is the number of slices and periodic boundary conditions are assumed. For a Hamiltonian $H$ that is stoquastic in the basis $\mathcal{B}$, all the matrix elements of $e^{-\Delta\tau H}$ are nonnegative for any $\Delta\tau$, leading to nonnegative weights for each time slice. On the other hand, non-stoquastic Hamiltonians, whose local terms have positive off-diagonal entries, induce negative weights and generally lead to the sign problem[1,6] unless certain symmetries are present.

The concept of stoquasticity is also important from a computational complexity-theory viewpoint. For example, the complexity class StoqMA associated with the problem of deciding whether the ground-state energy of stoquastic local Hamiltonians is above or below certain values, is expected to be strictly contained in the complexity class QMA, that poses the same decision problem for general local Hamiltonians[4]. In addition, StoqMA appears as an essential part of the complexity classification of two-local qubit Hamiltonian problem[7].

However, stoquasticity does not imply efficient (i.e., polynomial-time) equilibration. For example, finding the ground-state energy of a classical Ising model—which is trivially stoquastic—is already NP-hard[8]. Conversely, non-stoquasticity does not imply inefficiency: there exist numerous cases where an apparent sign problem (i.e., non-stoquasticity) is the result of a naive basis choice that can be transformed away, resulting in efficient equilibration[7,9–11]. Here, we focus on the latter, i.e., whether non-stoquasticity can be "cured".

To this end, we first propose an alternative definition of stoquasticity that is based on the computational complexity associated with transforming non-stoquastic Hamiltonians into stoquastic ones. Then, we proceed by proving that finding such a transformation for general local Hamiltonians, even if restricted to the single-qubit Clifford group or the single-qubit orthogonal group, is computationally hard. Along the way, we provide several results of independent interest, in particular an algorithm to efficiently group local Hamiltonian terms, and an algorithm to efficiently decide the curing problem using Pauli operators. We conclude by discussing some implications of our results, employing planted solution ideas, and also some potential cryptographic applications.

## Results

**Computationally stoquastic Hamiltonians.** To motivate our alternative definition, we first note that any Hamiltonian can trivially be presented as stoquastic via diagonalization. However, the complexity of finding the diagonalizing basis generally grows exponentially with the size of the system (as noted in ref. [6]) and the new basis will generally be highly nonlocal and hence not efficiently representable. We also note that it is straightforward to construct examples where apparent non-stoquasticity may be transformed away. For example, consider the $n$-spin Hamiltonian $H_{XZ} = \sum \tilde{J}_{ij} X_i X_j - \sum J_{ij} Z_i Z_j$ with nonnegative $J_{ij}, \tilde{J}_{ij}$, where $X_i$ and $Z_i$ are the Pauli matrices acting on spin $i$. This Hamiltonian is non-stoquastic, but can easily be converted into a stoquastic form. Denoting the Hadamard gate (which swaps $X$ and $Z$) by $W$, consider the transformed Hamiltonian $H_{ZX} = W^{\otimes n} H_{XZ} W^{\otimes n} = -\sum \tilde{J}_{ij} X_i X_j + \sum J_{ij} Z_i Z_j$, which is stoquastic. The sign problem of the original Hamiltonian, $H_{XZ}$, can thus be efficiently cured by a unit-depth circuit of single-qubit rotations. Moreover, thermal averages are invariant under unitary transformations; namely, defining $\langle A \rangle_H \equiv \frac{\mathrm{Tr}(e^{-\beta H} A)}{\mathrm{Tr}(e^{-\beta H})}$, it is straightforward to check that $\langle A \rangle_H = \langle UAU^\dagger \rangle_{UHU^\dagger}$. Therefore, if QMC is run on the transformed, stoquastic Hamiltonian, it is no longer slowed down by the sign problem. Finally, note that Definition 1 implies that a local Hamiltonian, $H = \sum_{a=1}^M H_a$, is stoquastic if all terms $H_a$ are stoquastic. However, there always remains some arbitrariness in the manner in which the total Hamiltonian is decomposed into the various terms. Consider, e.g., $H = -2X_1 + X_1 Z_2$. The second term separately is non-stoquastic, whereas the sum is stoquastic. This suggests that the grouping of terms matters (see the Methods section, "Grouping terms without changing the basis").

The above considerations motivate a reexamination of the concept of stoquasticity from a complexity-theory perspective, which can have important consequences for QMC simulations. (A related approach was discussed in ref. [12].) For example, given a $k$-local non-stoquastic Hamiltonian $H = \sum_a H_a$ (where each summand is a $k$-local term, i.e., a tensor product of at most $k$ non-identity single-qubit Pauli operators), similarly to ref. [13], we may ask whether there exists a constant-depth quantum circuit $U$ such that $H' = UHU^\dagger$ can be written as a $k'$-local stoquastic Hamiltonian $H' = \sum_a H'_a$ and if so, what the complexity associated with finding it is. It is the answer to the latter question that determines whether the Hamiltonian in question should be considered computationally stoquastic, i.e., whether it is feasible (in a complexity theoretic sense) to find a "curing" transformation $U$, which would then allow QMC to compute thermal averages with $H$ by replacing it with $H'$. More formally, we propose the following definition:

**Definition 2** A *unitary transformation U "cures"* a non-stoquastic Hamiltonian $H$ (i.e., removes its sign problem) represented in a given basis if $H' = UHU^\dagger$ is stoquastic, i.e., its off-diagonal elements in the given basis are all non-positive. A

family of local Hamiltonians $\{H\}$ represented in a given basis is efficiently curable (or, equivalently, computationally stoquastic) if there exists a polynomial-time classical algorithm such that for any member of the family $H$, the algorithm can find a unitary $U$ and a Hamiltonian $H'$ with the property that $H' = UHU^{\dagger}$ is local and stoquastic in the given basis.

As an example, the Hamiltonian $H_{XZ}$ considered above is efficiently curable. General local Hamiltonians are unlikely to be efficiently curable as this would imply the implausible result that QMA=StoqMA[13].

Note that given some class of basis transformations, our definition distinguishes between the ability to cure a Hamiltonian efficiently or in principle. For example, deciding whether a Pauli group element $U = \otimes_{i=1}^{n} u_i$, where $u_i$ belongs to the single-qubit Pauli group $\mathcal{P}_1 = \{I, X, Y, Z\} \times \{\pm 1, \pm i\}$, can cure each term $\{H_a\}$ of a $k$-local Hamiltonian $H = \sum_a H_a$, which can be solved in polynomial time (see the Methods section, "Curing using Pauli operators"). However, the Hamiltonian $H = X_1 Z_2$ cannot be made stoquastic in principle using a Pauli group element, as conjugating it with Pauli operators results in $\pm X_1 Z_2$, both of which are non-stoquastic (see ref. [13] for the results on an intrinsic sign problem for local Hamiltonians). Therefore, the fact that general local Hamiltonians are not considered to be computationally stoquastic does not imply that curing is computationally hard for a given class of transformations.

The last example illustrates that, while the curing problem can be efficiently decided for the Pauli group, this group can cure a very limited family of Hamiltonians. This motivates us to consider the curing problem beyond the Pauli group. Our main result is a proof that even for particularly simple local transformations such as the single-qubit Clifford group and real-valued rotations, the problem of deciding whether a family of local Hamiltonians is curable cannot be solved efficiently, in the sense that it is equivalent to solving 3SAT and is hence NP-complete.

We assume that a $k$-local Hamiltonian $H = \sum_a H_a$ is described by specifying each of the local terms $H_a$, and the goal is to find a unitary $U$ that cures each of these local terms. In general, a unitary $U$ that cures the total Hamiltonian $H$ may not necessarily cure all $H_a$ separately. However, for all of the constructions in this paper, we prove that a unitary $U$ cures $H$ if and only if it cures all $H_a$ separately. The decomposition $\{H_a\}$ is merely used to guarantee that verification is efficient and the problem is contained in NP.

**Complexity of curing for the single-qubit Clifford group.** To study the computational complexity associated with finding a curing transformation $U$, we shall consider for simplicity single-qubit unitaries $U = \otimes_{i=1}^{n} u_i$ and only real-valued Hamiltonian matrices. As we shall show, even subject to these simplifying restrictions, the problem of finding a curing transformation $U$ is computationally hard when $U$ is not in the Pauli group.

We begin by considering the computational complexity of finding local rotations from a discrete and restricted set of rotations. Specifically, we consider the single-qubit Clifford group $\mathcal{C}_1$ (with group action defined as conjugation by one of its elements), defined as $\mathcal{C}_1 = \{U | U g U^{\dagger} \in \mathcal{P}_1 \; \forall g \in \mathcal{P}_1\}$, i.e., the normalizer of $\mathcal{P}_1$. It is well known that $\mathcal{C}_1$ is generated by $W$ and the phase gate $P = \text{diag}(1, i)$[14].

**Theorem 1** *Let $U = \otimes_{i=1}^{n} u_i$, where $u_i$ belongs to the single-qubit Clifford group. Deciding whether there exists a curing unitary $U$ for 3-local Hamiltonians is NP-complete.*

We prove this theorem by reducing the problem to the canonical NP-complete problem known as 3SAT (3-satisfiability)[15], beginning with the following lemma:

**Table 1 Mapping the problem of finding a suitable change of basis to the Boolean satisfiability problem**

| $x_i$ | $x_j$ | $x_k$ | $W(x)H_{ijk}^{(111)}W^{\dagger}(x)$ | $(\bar{x}_i \vee \bar{x}_j \vee \bar{x}_k)$ |
|-------|-------|-------|--------------------------------------|---------------------------------------------|
| 0 | 0 | 0 | Stoquastic | 1 |
| 0 | 0 | 1 | Stoquastic | 1 |
| 0 | 1 | 0 | Stoquastic | 1 |
| 1 | 0 | 0 | Stoquastic | 1 |
| 1 | 1 | 0 | Stoquastic | 1 |
| 1 | 0 | 1 | Stoquastic | 1 |
| 0 | 1 | 1 | Stoquastic | 1 |
| 1 | 1 | 1 | Non-stoquastic | 0 |

The Hamiltonian $W(x)H_{ijk}^{(111)}W^{\dagger}(x)$ is stoquastic (True) for any choice of the variables $x$ except $(x_i, x_j, x_k) = (1, 1, 1)$, which makes it non-stoquastic (False). This is precisely the truth table for the 3SAT clause $(\bar{x}_i \vee \bar{x}_j \vee \bar{x}_k)$.

**Lemma 1** *Let $u_i \in \{I, W\}$, where $I$ is the identity operation and $W$ is the Hadamard gate. Deciding whether there exists a curing unitary $U = \otimes_{i=1}^{n} u_i$ for 3-local Hamiltonians is NP-complete.*

To prove Lemma 1, we first introduce a mapping between 3SAT and 3-local Hamiltonians. Our goal is to find an assignment of $n$ binary variables $x_i \in \{0, 1\}$ such that the unitary $W(x) \equiv \otimes_{i=1}^{n} W_i^{x_i}$ [where $x \equiv (x_1, \ldots, x_n)$] rotates an input Hamiltonian to a stoquastic Hamiltonian. We use the following 3-local Hamiltonian as our building block:

$$H_{ijk}^{(111)} = Z_i Z_j Z_k - 3(Z_i + Z_j + Z_k) - (Z_i Z_j + Z_i Z_k + Z_j Z_k), \tag{1}$$

where $i$, $j$, and $k$ are three different qubit indices. It is straightforward to check that

$$W(x)H_{ijk}^{(111)}W^{\dagger}(x) = W_i^{x_i} \otimes W_j^{x_j} \otimes W_k^{x_k}(H_{ijk}^{(111)})W_i^{x_i} \otimes W_j^{x_j} \otimes W_k^{x_k} \tag{2}$$

is stoquastic ("True") for any combination of the binary variables $(x_i, x_j, x_k)$ except for $(1, 1, 1)$, which makes Eq. (2) non-stoquastic ("False").

This is precisely the truth table for the 3SAT clause $(\bar{x}_i \vee \bar{x}_j \vee \bar{x}_k)$, where $\vee$ denotes the logical disjunction and the bar denotes negation (see Table 1). We can define the other seven possible 3SAT clauses by conjugating $H_{ijk}^{(111)}$ with Hadamard or identity gates:

$$H_{ijk}^{(\alpha\beta\gamma)} = W_i^{\bar{\alpha}} \otimes W_j^{\bar{\beta}} \otimes W_k^{\bar{\gamma}} \cdot H_{ijk}^{(111)} \cdot W_i^{\bar{\alpha}} \otimes W_j^{\bar{\beta}} \otimes W_k^{\bar{\gamma}}. \tag{3}$$

The Hamiltonian $W(x)H_{ijk}^{(\alpha\beta\gamma)}W^{\dagger}(x)$ is non-stoquastic (corresponds to a clause that evaluates to False) only when $(x_i, x_j, x_k) = (\alpha, \beta, \gamma)$, and is stoquastic (True) for any other choice of the variables $x$. We have thus established a bijection between 3-local Hamiltonians $H_{ijk}^{(\alpha\beta\gamma)}$, with $(\alpha, \beta, \gamma) \in \{0, 1\}^3$, and the eight possible 3SAT clauses on three variables $(x_i, x_j, x_k) \in \{0, 1\}^3$. We denote these clauses, which evaluate to False iff $(x_i, x_j, x_k) = (\alpha, \beta, \gamma)$, by $C_{ijk}^{(\alpha\beta\gamma)}$.

The final step of the construction is to add together such "3SAT-clause Hamiltonians" to form

$$H_{3SAT} = \sum_C H_{ijk}^{(\alpha\beta\gamma)}, \tag{4}$$

where $C$ is the set of all $M$ clauses in the given 3SAT instance $\wedge C_{ijk}^{(\alpha\beta\gamma)}$. Having established a bijection between 3SAT clauses and 3SAT-clause Hamiltonians, the final step is to show that finding $x$

such that

$$H' = W(x)H_{3SAT}W(x) \qquad (5)$$

is stoquastic for every $H_{3SAT}$, is equivalent to solving the NP-complete problem of finding satisfying assignments $x$ for the corresponding 3SAT instances. To prove the equivalence, we show (i) that satisfying a 3SAT instance implies that the corresponding $H_{3SAT}$ is cured, and (ii) that if $H_{3SAT}$ is cured, this implies that the corresponding 3SAT instance is satisfied.

(i)   Note that any assignment $x$ that satisfies the given 3SAT instance also satisfies each individual clause. It follows from the bijection we have established that such an assignment cures each corresponding 3SAT-clause Hamiltonian individually. The stoquasticity of $H'$ then follows by noting that the tensor product of a stoquastic Hamiltonian with the identity matrix is still stoquastic and the sum of stoquastic Hamiltonians is stoquastic.

(ii)  To simplify the argument, we assume that each clause has exactly three variables. This version of 3SAT, sometimes called EXACT-3SAT, remains NP-hard[15]. We prove that an unsatisfied 3SAT instance implies that the corresponding $H_{3SAT}$ is not cured. It suffices to focus on a particular clause $C_{ijk}^{(\alpha\beta\gamma)}$. The choice of variables that makes this clause False rotates the corresponding 3SAT-clause Hamiltonian to one that contains a non-stoquastic $+X_iX_jX_j$ term, which generates positive off-diagonal elements in specific locations in the matrix representation of $H_{3SAT}$. In what follows, we show that no other 3SAT-clause Hamiltonian in $H_{3SAT}$ contains a $\pm X_iX_jX_k$ term and therefore these positive off-diagonal elements cannot be canceled out or made negative regardless of the choice of the other variables in the assignment. To see this, we first note that a 3SAT-clause Hamiltonian that does not contain $x_i$, $x_j$, and $x_k$, cannot generate a $\pm X_iX_jX_k$ term. Second, a choice of assignment for $x_i$, $x_j$, and $x_k$ that does not satisfy $C_{ijk}^{(\alpha\beta\gamma)}$ would satisfy any other 3SAT clause on these three variables. A satisfied 3SAT clause also does not generate an $X_iX_jX_k$ term. Therefore, the rotated Hamiltonian is guaranteed to be non-stoquastic.

This establishes that the problem is NP-Hard. Checking whether a given $U$ cures all the local terms $\{H_a\}$ is efficient and therefore the problem is NP-complete.

To complete the proof of Theorem 1, let us consider the modified Hamiltonian

$$\tilde{H}_{3SAT} = H_{3SAT} + cH_0, \quad H_0 = -\sum_{i=1}^{n}(X_i + Z_i), \qquad (6)$$

where $H_0$ is manifestly stoquastic and $c$ is any number larger than the maximum number of clauses that any variable appears in. As there are $M$ clauses, we simply choose $c = O(1)M$, which is still a polynomial in the number of variables. (Note that even a restricted variant of 3SAT with each variable restricted to appear at most in a constant number of clauses is still NP-Complete[16] and therefore we can take $c$ to be a constant.) The goal is to find a unitary $U = \otimes_{i=1}^{n} u_i$ with $u_i \in \mathcal{C}_1$ that cures $\tilde{H}_{3SAT}$. Note first that any choice of $U = \otimes_{i=1}^{n} W_i^{x_i}$ that cures $H_{3SAT}$ also cures $\tilde{H}_{3SAT}$. Second, note that choosing any $u_i$ that keeps $H_0$ stoquastic is equivalent to choosing one of the elements of $\mathcal{C}_1' \equiv \{I, X, W, XW\} \subset \mathcal{C}_1$ (e.g., the phase gate, which is an element of $\mathcal{C}_1$, maps $X$ to $Y$ so it is excluded, as is $WX$, which maps $Z$ to $-X$). Therefore, by choosing $c$ to be large enough, any choice of $u_i \in \mathcal{C}_1 \backslash \mathcal{C}_1'$ would transform $\tilde{H}_{3SAT}$ into a non-stoquastic Hamiltonian. It follows that if $u_i \in \mathcal{C}_1$ and is to cure $\tilde{H}_{3SAT}$ then in fact it must be an element of $\mathcal{C}_1'$.

Next, we note that conjugating a matrix by a tensor product of $X$ or identity operators only shuffles the off-diagonal elements, but never changes their values (for a proof see the Methods section, "Conjugation by a product of $X$ operators"). Therefore, for the purpose of curing a Hamiltonian, applying $X$ is equivalent to applying $I$ and applying $XW$ is equivalent to applying $W$. With this observation, the set of operators that can cure a Hamiltonian is effectively reduced from $\{I, W, XW, X\}$ to $\{I, W\}$. According to Lemma 1, deciding whether such a curing transformation exists is NP-complete.

**Complexity of curing for the single-qubit orthogonal group.** Similarly, we can use Lemma 1 to show that the problem of curing the sign problem remains NP-complete when the set of allowed rotations is extended to the continuous group of single-qubit orthogonal matrices, i.e., transformations of the form $Q = \otimes_{i=1}^{n} q_i$, where $q_i^T q_i = I \; \forall i$. Namely:

**Theorem 2** *Deciding whether there exists a curing orthogonal transformation* Q *for 6-local Hamiltonians is NP-complete.*

See the Methods section, "Proof of Theorem 2", for the proof. In analogy to the proof of Theorem 1, the crucial step is to show that by promoting each $Z$, $X$, and $W$ to a two-qubit operator, the continuous set of possible curing transformations reduces to the discrete set considered in Lemma 1.

**Implications and applications.** An immediate and striking implication of Theorem 1 is that even under the promise that a non-stoquastic Hamiltonian can be cured by one-local Clifford unitaries (corresponding to trivial basis changes), the problem of actually finding this transformation is unlikely to have a polynomial-time solution.

An interesting implication of Theorem 2 is the possibility of constructing "secretly stoquastic" Hamiltonians. That is, one may generate stoquastic quantum many-body Hamiltonians $H_{stoq}$, but present these in a "scrambled" non-stoquastic form $H_{nonstoq} = UH_{stoq}U^{\dagger}$, where $U$ is a tensor product of single-qubit orthogonal matrices (or in the general case a constant-depth quantum circuit). We conjecture that the latter Hamiltonians will be computationally hard to simulate using QMC by parties that have no access to the "descrambling" circuit $U$. In other words, it is possible to generate efficiently simulable spin models that might be inefficient to simulate unless one has access

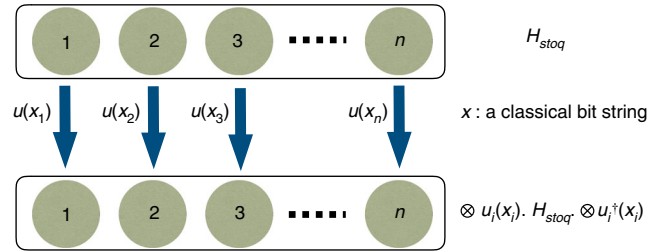

**Fig. 1** A classical bit string connecting a stoquastic Hamiltonian to a seemingly non-stoquastic Hamiltonian. One may generate an $n$-qubit stoquastic Hamiltonian $H_{stoq}$ and then transform it using a randomly chosen unitary (specified by a classical bit string) to bring it into a seemingly non-stoquastic form. Unlike the generated stoquastic Hamiltonian, the simulation of the seemingly non-stoquastic Hamiltonian can be computationally hard. Also, as discussed here, given a non-stoquastic Hamiltonian, finding the bit string that converts it into a stoquastic Hamiltonian can be computationally hard in general. Therefore, the classical bit string can serve as a secret key, without which certain properties of $H_{stoq}$ cannot be efficiently simulated

to the "secret key" to make them stoquastic (see Fig. 1). This observation may potentially have cryptographic applications (see the Methods section, "Encryption based on secretly stoquastic Hamiltonians").

Our work also has implications for the connection between the sign problem and the NP-hardness of a QMC simulation. A prevailing view of this issue associates the origin of the NP-hardness of a QMC simulation to the relation between a ("fermionic") Hamiltonian that suffers from a sign problem and the corresponding ("bosonic") Hamiltonian obtained by replacing every coupling coefficient by its absolute value. Consider the following example:. $H_X = \sum_{ij} J_{ij} X_i X_j$, with $J_{ij}$ randomly chosen from the set $\{0, \pm J\}$ on a three-dimensional lattice, has a sign problem. Deciding whether its ground-state energy is below a given bound is NP-complete[8]. Deciding the same for its bosonic and sign-problem-free version $H_{|X|} = \sum_{ij} |J_{ij}| X_i X_j$ is in BPP (classical polynomial time with bounded error) since this Hamiltonian is that of a simple ferromagnet. The conclusion drawn in ref. [6] was that since the bosonic version is easy to simulate, the sign problem is the origin of the NP-hardness of a QMC simulation of this model ($H_X$).

The view we advocate here is that a solution to the sign problem is to find an efficiently computable curing transformation that removes it in such a way that the model has the same physics (in general the fermionic and bosonic versions of the same Hamiltonian do not), i.e., conserves thermal averages. In the above example, computing thermal averages via a QMC simulation of $H_X$ is the same as for $H_Z = W^{\otimes n} H_X W^{\otimes n} = \sum_{ij} J_{ij} Z_i Z_j$, which is stoquastic. Thus, the sign problem of $H_X$ is efficiently curable, after which (when it is presented as $H_Z$) deciding its ground-state energy remains NP-hard.

## Discussion

We have proposed an alternative definition of stoquasticity (or absence of the sign problem) of quantum many-body Hamiltonians that is motivated by computational complexity considerations. We discussed the circumstances under which non-stoquastic Hamiltonians can in fact be made stoquastic by the application of single-qubit rotations and in turn potentially become efficiently simulable by QMC algorithms. We have demonstrated that finding the required rotations is computationally hard when they are restricted to the one-qubit Clifford group or one-qubit continuous orthogonal matrices.

These results raise multiple questions of interest. It is important to clarify the computational complexity of finding the curing transformation in the case of constant-depth circuits that also allow two-body rotations, whether discrete or continuous. Also, since our NP-completeness proof involved 3- and 6-local Hamiltonians, it is interesting to try to reduce it to 2-local building blocks. Another direction into which these results can be extended is to relax the constraints on the off-diagonal elements and require that they are smaller than some small $\varepsilon > 0$. This is relevant when some small positive off-diagonal elements can be ignored in a QMC simulation.

Finally, it is natural to reconsider our results from the perspective of quantum computing. Namely, for non-stoquastic Hamiltonians that are curable, do there exist quantum algorithms that cure the sign problem more efficiently than is possible classically? With the advent of quantum computers, specifically quantum annealers, it may be the case that these can be used as quantum simulators, and as such they will not be plagued by the sign problem. Will such physical implementations of quantum computers offer advantages over classical computing even for

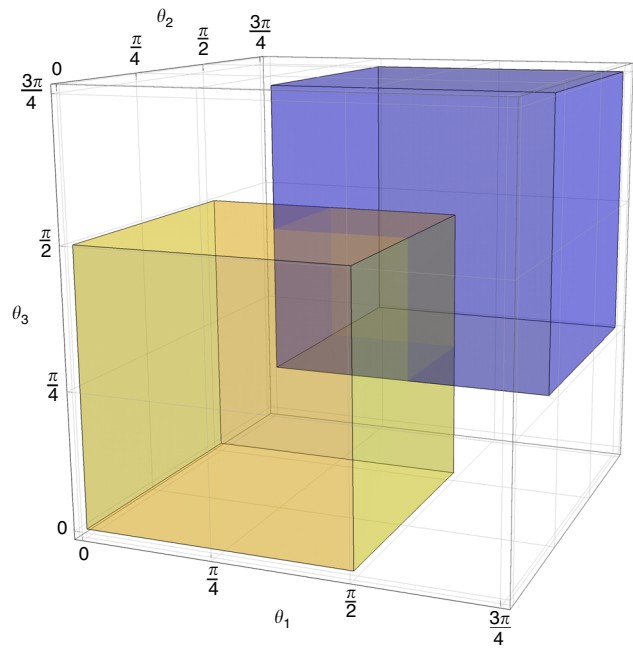

**Fig. 2** Orthogonal rotations that can cure the clause Hamiltonians introduced in Eq. (2). The yellow region depicts the angles of rotations that can cure $H_{123}^{(111)}$ and the blue region depicts the angles of the curing rotations for $H_{123}^{(000)}$

problems that are incurably non-stoquastic? We leave these as open questions to be addressed in future studies.

## Methods
**Proof of Theorem 2**. The proof builds on that of Theorem 1, but first we note that the clause Hamiltonians introduced in Eq. (2) can have curing solutions that are orthogonal rotations outside the Clifford group (see Fig. 2). To deal with this richer set of rotations—which is now a continuous group—we promote each $Z$, $X$, and $W$ in the clause Hamiltonians to a two-qubit operator: $Z_i \mapsto \bar{Z}_i \equiv Z_{2i-1} Z_{2i}$, $X_i \mapsto \bar{X}_i \equiv X_{2i-1} X_{2i}$, $W_i^\alpha \mapsto \overline{W}_i^\alpha \equiv W_{2i-1}^\alpha \otimes W_{2i}^\alpha$. Thus Eq. (1) becomes

$$\bar{H}_{ijk}^{(111)} = \bar{Z}_i \bar{Z}_j \bar{Z}_k - 3(\bar{Z}_i + \bar{Z}_j + \bar{Z}_k) - (\bar{Z}_i \bar{Z}_j + \bar{Z}_i \bar{Z}_k + \bar{Z}_j \bar{Z}_k). \quad (7)$$

Let $\bar{W}(x) \equiv \otimes_{i=1}^n \bar{W}_i^{x_i}$. It is again straightforward to check that $\bar{W}(x) \bar{H}_{ijk}^{(111)} \bar{W}^\dagger(x)$ is stoquastic for any combination of the binary variables $(x_i, x_j, x_k)$ except for $(1, 1, 1)$. Likewise, generalizing Eq. (3), we define

$$\bar{H}_{ijk}^{(\alpha\beta\gamma)} = \bar{W}_i^{\bar{\alpha}} \otimes \bar{W}_j^{\bar{\beta}} \otimes \bar{W}_k^{\bar{\gamma}} \cdot \bar{H}_{ijk}^{(111)} \cdot \bar{W}_i^{\bar{\alpha}} \otimes \bar{W}_j^{\bar{\beta}} \otimes \bar{W}_k^{\bar{\gamma}}. \quad (8)$$

$\bar{H}_{ijk}^{(\alpha\beta\gamma)}$ is a clause Hamiltonian corresponding to a clause in the 3SAT instance. Similarly, $\bar{W}(x) \bar{H}_{ijk}^{(\alpha\beta\gamma)} \bar{W}(x)$ is stoquastic for any combination of the binary variables $(x_i, x_j, x_k)$ except for $(\alpha, \beta, \gamma)$. Generalizing Eq. (4) we define

$$\bar{H}_{3SAT} = \sum_C \bar{H}_{ijk}^{(\alpha\beta\gamma)}, \quad (9)$$

where $C$ denotes the corresponding set of clauses in a 3SAT instance, constructed just as in the proof of Lemma 1. This, again, establishes a bijection between 3SAT clauses and "3SAT-clause Hamiltonians", now of the form $\bar{H}_{3SAT}$.

In lieu of Eq. (4), we consider the 6-local Hamiltonian $\widetilde{H}_{3SAT}$

$$\widetilde{H}_{3SAT} = \bar{H}_{3SAT} + c H_0', \quad H_0' = -\sum_{i=1}^n 2\bar{Z}_i + \bar{X}_i, \quad (10)$$

where $c = O(1)$. Just as in the proof of Theorem (1), we prove that (i) any satisfying assignment of a 3SAT instance provides a curing $Q$ for the corresponding Hamiltonian $\widetilde{H}_{3SAT}$, and (ii) any $Q$ that cures $\widetilde{H}_{3SAT}$ provides a satisfying assignment for the corresponding 3SAT instance. Because of the relation between a single-qubit orthogonal matrix and a single-qubit rotation, it suffices to prove the hardness only for pure rotations (see the next subsection for the relation between a

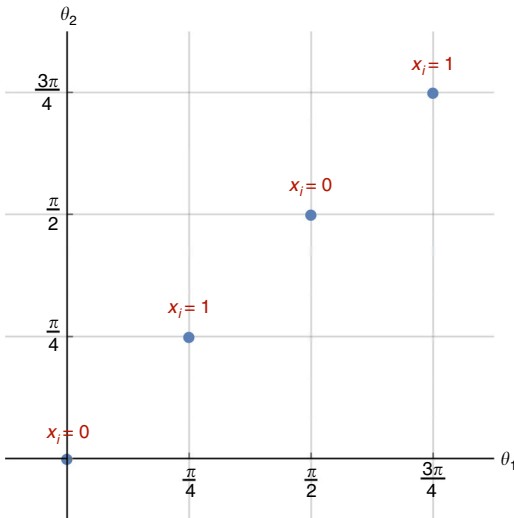

**Fig. 3** Reducing the continuous region of solutions to a discrete set. The set of possible orthogonal transformations that can cure $\widetilde{H}_{3SAT}$ (introduced in Eq. (10)) reduces to a discrete set. We assign the value of each binary variable satisfying a 3SAT instance depending on the value of the curing transformation. We set $x_i = 0$ if the curing transformation has $(\theta_{2i-1}, \theta_{2i}) \in \left\{ (0, 0), (\frac{\pi}{2}, \frac{\pi}{2}) \right\}$, while we assign $x_i = 1$ if the curing transformation has $(\theta_{2i-1}, \theta_{2i}) \in \left\{ \left(-\frac{\pi}{4}, -\frac{\pi}{4}\right), \left(\frac{\pi}{4}, \frac{\pi}{4}\right) \right\}$

single-qubit real–orthogonal matrix and a single-qubit rotation); we let $R(\theta_i) = \begin{bmatrix} \cos\theta_i & -\sin\theta_i \\ \sin\theta_i & \cos\theta_i \end{bmatrix}$ denote a rotation by angle $\theta_i$.

(i) Let $P(x)$ denote the product of $2n$ single-qubit rotations such that if $x_i = 0$ then qubits $2i - 1$ and $2i$ are unchanged, or if $x_i = 1$ then they are both rotated by $R(\frac{\pi}{4})$:

$$P(x) \equiv \otimes_{i=1}^{n} \left( R\left(\frac{\pi}{4}\right)^{x_i} \otimes R\left(\frac{\pi}{4}\right)^{x_i} \right), \tag{11}$$

where $x$ is a $n$-bit string $x = (x_1, …, x_n)$. Let us now show that if the 3SAT instance has a satisfying assignment $x^*$ then $P(x^*)\widetilde{H}_{3SAT}P^T(x^*)$ is stoquastic. Note that $R(\frac{\pi}{4}) = XW$, and as we discussed in the article (under "Complexity of curing for the single-qubit Clifford group") it is equivalent to $W$ for curing.

To prove the claim, note that $x^*$ necessarily satisfies each individual clause of the 3SAT instance, and therefore makes the corresponding clause Hamiltonian stoquastic, i.e., $P(x^*)\bar{H}_{ijk}^{(\alpha\beta\gamma)}P^T(x^*)$ is stoquastic $\forall C_{ijk}^{(\alpha\beta\gamma)}$. Also, $P(x)H_0'P^T(x)$ is clearly stoquastic for any $x$, where $H_0' = -\sum_{i=1}^{n} 2\bar{Z}_i + \bar{X}_i$ [Eq. (10)]. The stoquasticity of $P(x^*)\widetilde{H}_{3SAT}P^T(x^*)$ then follows immediately.

(ii) We need to prove that any rotation that cures $\bar{H}_{3SAT}$ provides a satisfying assignment for the corresponding 3SAT instance. We do this in two steps:

(a) Below, under "A useful lemma", we prove that for any $\widetilde{H}_{3SAT}$, any curing rotation $R = \otimes_{i=1}^{2n} R(\theta_i)$ has to satisfy the condition that $(\theta_{2i-1}, \theta_{2i}) \in \left\{ (\frac{\pi}{2}, \frac{\pi}{2}), (\frac{\pi}{4}, \frac{\pi}{4}), (0, 0), (\frac{-\pi}{4}, \frac{-\pi}{4}) \right\}$ $\forall i$. This is the crucial step, since it reduces the problem from a continuum of angles to a discrete set. If $(\theta_{2i-1}, \theta_{2i}) \in \left\{ (0, 0), (\frac{\pi}{2}, \frac{\pi}{2}) \right\}$, we assign $x_i = 0$, while if $(\theta_{2i-1}, \theta_{2i}) \in \left\{ \left(-\frac{\pi}{4}, -\frac{\pi}{4}\right), \left(\frac{\pi}{4}, \frac{\pi}{4}\right) \right\}$, we assign $x_i = 1$, since rotations with the angles in each pair have the same effect.

(b) If $R$ cures $\bar{H}_{3SAT}$, $x = \{x_i\}$ satisfies the corresponding 3SAT instance. See Fig. 3 for the discrete set of solutions and the corresponding assignments.

To see this, we first note that if $R$ cures $\widetilde{H}_{3SAT}$ it must cure all the clauses separately: Using step (a), we know that any such solution must be one of the four possible cases. Therefore, if $R$ were to cure $\widetilde{H}_{3SAT}$ but does not cure one of the 3SAT-clause Hamiltonians, it would result in a $\bar{X}_i\bar{X}_j\bar{X}_k$ term in the corresponding clause. Since no other 3SAT-clause Hamiltonian in $\widetilde{H}_{3SAT}$ contains an identical $\bar{X}_i\bar{X}_j\bar{X}_k$ term, these positive off-diagonal elements cannot be canceled out or made negative, regardless of the choice of the other variables in the assignment. Therefore, if $R$ cures $\widetilde{H}_{3SAT}$, it also necessarily separately cures all the terms in $\widetilde{H}_{3SAT}$. By construction, if $R$ cures a term $\bar{H}_{ijk}^{(\alpha\beta\gamma)}$, the string $x$ satisfies the

corresponding 3SAT clause $C_{ijk}^{(\alpha\beta\gamma)}$. Thus $x$ satisfies all the clauses in the corresponding 3SAT instance. The decision problem for the existence of $R$ (and hence $Q$) is therefore NP-hard. Given a unitary $U$ and a set of local terms $\{H_a\}$, verifying whether $U$ cures all of the terms is clearly efficient and therefore this problem is NP-complete.

**Relation between an orthogonal matrix and a rotation**. The condition $q_i q_i^T = I$ forces each real–orthogonal matrix $q_i$ to be either a reflection or a rotation of the form

$$q_i = \begin{bmatrix} \cos\theta_i & a_i \sin\theta_i \\ \sin\theta_i & -a_i \cos\theta_i \end{bmatrix} \tag{12}$$

with $a_i = +1$ (a reflection) or $a_i = -1$ (a rotation). The operators $X$, $Z$, and Hadamard, are included in the family with $a_i = 1$; $I$ and $iY = XZ$ are in the family with $a_i = -1$. Note that $\forall H, \forall \theta_i : q_i(\theta_i)Hq_i^T(\theta_i) = q_i(\theta_i + \pi)Hq_i^T(\theta_i + \pi)$. Therefore, the angles that cure a Hamiltonian are periodic with a period of $\pi$. Hence, it suffices to consider the curing solutions only in one period: $\theta_i \in \left( \frac{-\pi}{2}, \frac{+\pi}{2} \right]$.

Next, observe that a reflection by angle $\theta_i$ can be written as

$$\begin{bmatrix} \cos\theta_i & \sin\theta_i \\ \sin\theta_i & -\cos\theta_i \end{bmatrix} = X \begin{bmatrix} \cos\frac{\pi}{2} - \theta_i & -\sin\frac{\pi}{2} - \theta_i \\ \sin\frac{\pi}{2} - \theta_i & \cos\frac{\pi}{2} - \theta_i \end{bmatrix} = XR\left(\frac{\pi}{2} - \theta_i\right), \tag{13}$$

where

$$R(\theta_i) = \begin{bmatrix} \cos\theta_i & -\sin\theta_i \\ \sin\theta_i & \cos\theta_i \end{bmatrix}. \tag{14}$$

As discussed below (under "Conjugation by a product of $X$ operators"), if $XR\left(\frac{\pi}{2} - \theta_i\right)$ is a curing operator so is $R\left(\frac{\pi}{2} - \theta_i\right)$. Therefore, any curing $Q = \otimes_{i=1}^{n} q_i$ provides a curing $R = \otimes_{i=1}^{n} R(\theta_i)$. Hence, the NP-completeness of the decision problem for $R$ implies the NP-hardness of the decision problem for $Q$, which is the statement of Theorem 2.

**A useful lemma**. Here, we prove that any curing rotation $R = \otimes_{i=1}^{2n} R(\theta_i)$ for any $\widetilde{H}_{3SAT}$, as introduced in Eq. (10), must satisfy the condition that $(\theta_{2i-1}, \theta_{2i}) \in \left\{ (\frac{\pi}{2}, \frac{\pi}{2}), (\frac{\pi}{4}, \frac{\pi}{4}), (0, 0), (\frac{-\pi}{4}, \frac{-\pi}{4}) \right\}$ $\forall i$. In what follows, we show this for $i = 1$ (local rotations on the first two qubits), but the proof trivially works for any choice of $i$.

Our strategy is to expand any locally rotated $\widetilde{H}_{3SAT}$ on the first two qubits and then to find necessary conditions that any curing rotations on these two qubits must satisfy. With this motivation, we introduce the following lemma:

**Lemma 2** *Let* $\theta_1, \theta_2 \in \left(-\frac{\pi}{2}, \frac{\pi}{2}\right]$. *Consider the $2n$-qubit Hamiltonian*

$$H' = Z_1 \otimes Z_2 \otimes M_z + X_1 \otimes X_2 \otimes M_x + I_1 \otimes I_2 \otimes M_I, \tag{15}$$

*where $M_z$, $M_x$, and $M_I$ are Hamiltonians on $2n-2$ qubits satisfying the following two conditions*:

1. The absolute value of at least one element of $M_z$ is different from the absolute value of the corresponding element in $M_x$.
2. Both $M_x$ and $M_z$ have at least one negative element.

Then the only rotation $R(\theta_1) \otimes R(\theta_2)$ that can cure $H'$ has angles given by the following four points:

$$(\theta_1, \theta_2) \in \left\{ \left(\frac{\pi}{2}, \frac{\pi}{2}\right), \left(\frac{\pi}{4}, \frac{\pi}{4}\right), (0, 0), \left(\frac{-\pi}{4}, \frac{-\pi}{4}\right) \right\}. \tag{16}$$

To relate this lemma to our construction, note that any locally rotated $\widetilde{H}_{3SAT}$ is in the form of $H'$. To be more precise, we will choose $H' = R'\widetilde{H}_{3SAT}R'^T$, where $R' = \otimes_{i=3}^{2n} R(\theta_i)$.

We proceed by first proving the lemma. Then we show that both of the conditions introduced in the lemma are satisfied for our construction.

*Proof*. It is straightforward to check that

$$R(\theta_i)XR(-\theta_i) = \cos 2\theta_i X - \sin 2\theta_i Z, \tag{17}$$

$$R(\theta_i)ZR(-\theta_i) = \sin 2\theta_i X + \cos 2\theta_i Z. \tag{18}$$

Using this we have

$$\begin{aligned} H'' &= [R(\theta_1) \otimes R(\theta_2)]H'[R(-\theta_1) \otimes R(-\theta_2)] \\ &= X_1 \otimes X_2 \otimes [\sin 2\theta_1 \sin 2\theta_2 M_z + \cos 2\theta_1 \cos 2\theta_2 M_x] \\ &\quad + X_1 \otimes Z_2 \otimes [\sin 2\theta_1 \cos 2\theta_2 M_z - \cos 2\theta_1 \sin 2\theta_2 M_x] \\ &\quad + Z_1 \otimes X_2 \otimes [\cos 2\theta_1 \sin 2\theta_2 M_z - \sin 2\theta_1 \cos 2\theta_2 M_x] \\ &\quad + Z_1 \otimes Z_2 \otimes [\cos 2\theta_1 \cos 2\theta_2 M_z + \sin 2\theta_1 \sin 2\theta_2 M_x] \\ &\quad + I_1 \otimes I_2 \otimes M_I. \end{aligned} \tag{19}$$

We next find necessary conditions that $\theta_1$ and $\theta_2$ must satisfy in order to make $H''$ stoquastic.

Let $A$ to $E$ be arbitrary matrices, and let [0] denote the all-zero matrix. We note that $H''$ can be stoquastic only if the $X_1Z_2 \otimes B$ term is zero (where we have dropped the tensor product between the first two qubits for notational simplicity). To see this, we first note that the matrix $X_1Z_2 \otimes B$ has both $+B$ and $-B$ as distinct

off-diagonal elements for any nonzero matrix $B$ (a similar observation holds for $Z_1 X_2 \otimes C$). Second, we note that there are no common off-diagonal elements between $X_1 X_2 \otimes A$, $X_1 Z_2 \otimes B$, and $Z_1 X_2 \otimes C$. Third, there is no common off-diagonal elements between these three matrices and $Z_1 Z_2 \otimes D$ and $I_1 I_2 \otimes E$. Therefore, these terms cannot make the $\pm B$ in $X_1 Z_2 \otimes B$ non-positive.

Thus, for $H''$ to become stoquastic it is necessary to have $B = [0]$:

$$\sin 2\theta_1 \cos 2\theta_2 M_z - \cos 2\theta_1 \sin 2\theta_2 M_x = [0]. \quad (20)$$

Similar reasoning for $Z_1 X_2 \otimes C$ yields

$$\cos 2\theta_1 \sin 2\theta_2 M_z - \sin 2\theta_1 \cos 2\theta_2 M_x = [0]. \quad (21)$$

If the absolute value of at least one element of $M_z$ is different from the absolute value of the corresponding element in $M_x$ (i.e., Condition 1 is satisfied), comparing the corresponding expressions from Eqs. (20) and (21), we can conclude that

$$\sin 2\theta_1 \cos 2\theta_2 = \cos 2\theta_1 \sin 2\theta_2 = 0. \quad (22)$$

Equation (22) gives rise to two possible cases: (i) $\sin 2\theta_1 = \sin 2\theta_2 = 0$, or (ii) $\cos 2\theta_1 = \cos 2\theta_2 = 0$. For $\theta_1, \theta_2 \in \left(-\frac{\pi}{2}, \frac{\pi}{2}\right]$. These are the eight possible solutions:

$$(\theta_1, \theta_2) \in \left\{0, \frac{\pi}{2}\right\} \times \left\{0, \frac{\pi}{2}\right\}, \text{ or } (\theta_1, \theta_2) \in \left\{\pm\frac{\pi}{4}\right\} \times \left\{\pm\frac{\pi}{4}\right\}. \quad (23)$$

Now, we observe that Condition 2 generates additional constraints on the allowed values of $\theta_1$ and $\theta_2$. To see this, we consider the $X_1 X_2 \otimes A$ term in Eq. (19) in the two possible cases. For case (i), when $\sin 2\theta_1 = \sin 2\theta_2 = 0$ and hence $\cos 2\theta_1 = \pm 1$ and $\cos 2\theta_2 = \pm 1$, this term becomes $X_1 X_2 \otimes \cos 2\theta_1 \cos 2\theta_2 M_x$. If $M_x$ has any negative element, then the two combinations such that $\cos 2\theta_1 \cos 2\theta_2 = -1$ flip the sign of this element and make the term non-stoquastic. That is, if $M_x$ has any negative element, the only rotations that can keep $H''$ stoquastic satisfy $\cos 2\theta_1 = \cos 2\theta_2 = 1$ or $\cos 2\theta_1 = \cos 2\theta_2 = -1$. Similarly, for case (ii), when $\cos 2\theta_1 = \cos 2\theta_2 = 0$, if $M_z$ has any negative elements, the only rotations that can keep $H''$ stoquastic satisfy $\sin 2\theta_1 = \sin 2\theta_2 = 1$ or $\sin 2\theta_1 = \sin 2\theta_2 = -1$.

To summarize, if both conditions hold, the solutions are necessarily one of these four points:

$$\sin(2\theta_1) = 0, \cos(2\theta_1) = 1, \sin(2\theta_2) = 0, \cos(2\theta_2) = 1$$
$$\Rightarrow (\theta_1, \theta_2) = (0, 0) \quad (24)$$

$$\sin(2\theta_1) = 0, \cos(2\theta_1) = -1, \sin(2\theta_2) = 0, \cos(2\theta_2) = -1$$
$$\Rightarrow (\theta_1, \theta_2) = \left(\frac{\pi}{2}, \frac{\pi}{2}\right) \quad (25)$$

$$\sin(2\theta_1) = 1, \cos(2\theta_1) = 0, \sin(2\theta_2) = 1, \cos(2\theta_2) = 0$$
$$\Rightarrow (\theta_1, \theta_2) = \left(\frac{\pi}{4}, \frac{\pi}{4}\right) \quad (26)$$

$$\sin(2\theta_1) = -1, \cos(2\theta_1) = 0, \sin(2\theta_2) = -1, \cos(2\theta_2) = 0$$
$$\Rightarrow (\theta_1, \theta_2) = \left(\frac{-\pi}{4}, \frac{-\pi}{4}\right) \quad (27)$$

Having proved the lemma, we now proceed with identifying the properties of $M_z$ and $M_x$ for our construction. As mentioned earlier, we choose $H' = R' \tilde{H}_{3\text{SAT}} R'^T$, where $R' = \otimes_{i=3}^{2n} R(\theta_i)$.

$M_z$ can be written as $H_z - 2I$, where $H_z$ denotes the terms in $M_z$ coming from $\bar{H}_{3\text{SAT}}$. Similarly, $M_x = H_x - I$, where $H_x$ denotes the terms in $M_x$ coming from $\bar{H}_{3\text{SAT}}$. The term $\bar{Z}_1 \otimes H_z$ (recall that $\bar{Z}_i \equiv Z_{2i-1} Z_{2i}$ and $\bar{X}_i \equiv X_{2i-1} X_{2i}$) is composed of rotated 3SAT-clause Hamiltonians that share $\bar{x}_1$ in their corresponding 3SAT clauses. Therefore, we have

$$H_z = R' \left( \sum_{C_{1jk}^{(1\beta\gamma)} \in C} \bar{W}_j^{\bar{\beta}} \otimes \bar{W}_k^{\bar{\gamma}} (\bar{Z}_j \bar{Z}_k - 3 - \bar{Z}_j - \bar{Z}_k) \bar{W}_j^{\bar{\beta}} \otimes \bar{W}_k^{\bar{\gamma}} \right) R'^T. \quad (28)$$

It is straightforward to check that each of the rotated 3SAT-clause Hamiltonians has only non-positive diagonal elements. Namely, using Eq. (17), it is straightforward to check that the max norm, defined as $\|A\|_{max} = max_{ij}|[A]_{ij}|$, of any rotated Pauli operator is at most 1, and therefore the same is true for any tensor product of rotated Pauli operators. In each 3SAT-clause Hamiltonian, there are three non-identity Pauli terms. Therefore, they cannot generate a diagonal element that is larger than 3. There is a $-3$ term for each clause, guaranteeing that all the diagonal terms remain non-positive.

As $H_z$ is a sum of these matrices with all non-positive diagonal elements, we conclude that all the diagonal elements of $H_z$ are non-positive. $H_x$ is similar to $H_z$, but with a sum over $C_{1jk}^{(0\beta\gamma)}$. Using similar arguments, we conclude that all the diagonal elements of $H_x$ are non-positive.

Therefore, all the diagonal elements of $M_x = H_x - I$ and $M_z = H_z - 2I$ are negative, and we conclude that Condition 2 is satisfied for our construction.

Now, we show that that the first condition also holds. Using the cyclic property of the trace and noting that all the terms in $H_z$ except the $-3$ are traceless, we have $\text{Tr}(H_z) = -3k$ with $k \in \mathbb{N}_0$ ($k = 0$ only if $H_z = 0$, i.e., when there is no $\bar{x}_1$ in any of the 3SAT clauses). Therefore we have $\text{Tr}(H_z - 2I) = -3k - 2^{2n-1}$. Using similar arguments, we conclude that $\text{Tr}(H_x) = -3k'$ with $k' \in \mathbb{N}_0$ and $\text{Tr}(H_x - I) = -3k' - 2^{2n-2}$ where $k' \in \mathbb{N}_0$. ($k' = 0$ only if $H_x = 0$, i.e., when there is no $x_1$ in any of the 3SAT clauses).

Clearly, the two traces cannot be equal for any value of $k$ and $k'$. From this, in addition to the already-established fact that all the diagonal elements of $H_x - I$ and $H_z - 2I$ are negative, we conclude that at least one diagonal element of $H_x - I$ is different from the corresponding element of $H_z - 2I$. Therefore, Condition 1 is also satisfied.

**Grouping terms without changing the basis.** As discussed here, one ambiguity in the definition of stoquastic Hamiltonians is in the choice of the set $\{H_a\}$. With this motivation, and ignoring the freedom in choosing a basis, we address the following question.

*Problem*: We are given the $k$-local $H = \sum_a H_a$, i.e., each $H_a$ acts nontrivially on at most $k$ qubits. In the same basis (without any rotation), find a new set $H'_a$ satisfying $H = \sum_a H'_a$, where each $H'_a$ is $k'$-local and stoquastic (if such a set exists).

Obviously, if the total Hamiltonian is stoquastic, then considering the total Hamiltonian as one single Hamiltonian is a valid solution with $k' = n$. This description of the Hamiltonian requires a $2^n \times 2^n$ matrix. We would prefer a $k'$-local Hamiltonian, i.e., a set consisting of a polynomial number of terms, each $2^{k'} \times 2^{k'}$, where $k'$ is a constant independent of $n$.

*Solution:* One simple strategy is to consider any $k'$-local combination of qubits, and to try to find a grouping that makes all of these $\binom{n}{k'}$ terms stoquastic. To do so, for any $k'$-local combination of qubits, we generate a set of inequalities. First, for a fixed combination of qubits, we add the terms in $H = \sum_a H_a$ that act nontrivially only on those $k'$ qubits, each with an unknown weight that will be determined later. Then we write down the conditions on the weights to ensure that all the off-diagonal elements are non-positive. This is done for all the $\binom{n}{k'}$ combinations to get the complete set of linear inequalities. By this procedure, the problem reduces to finding a feasible point for this set of linear inequalities, which can be solved efficiently. (In practice, one can use linear programming optimization tools to check whether such a feasible point exists.) When there is no feasible point for a specific value of $k'$, we can increase the value of $k'$ and search again.

*Example*: Assume we are given $H = Z_1 X_2 - 2X_2 + X_2 Z_3$ and the goal is to find a stoquastic description with $k' = 2$. We combine the terms acting on qubits 1 and 2 and then the terms acting on qubits 2 and 3 (there is no term on qubits 1 and 3). We construct $h_{1,2} = \alpha_1 Z_1 X_2 + \alpha_2(-2X_2)$ and $h_{2,3} = \alpha_3(-2X_2) + \alpha_4 X_2 Z_3$. There are two types of constraints: (1) constraints enforcing $H = h_{1,2} + h_{2,3}$:

$$\alpha_1 = \alpha_4 = 1, \alpha_2 + \alpha_3 = 1, \quad (29)$$

and (2) constraints from stoquasticity of each of the two Hamiltonians:

$$1 - 2\alpha_2 \leq 0, -1 - 2\alpha_2 \leq 0; \quad (30)$$

$$1 - 2\alpha_3 \leq 0, -1 - 2\alpha_3 \leq 0. \quad (31)$$

Simplifying these inequalities, we have $0.5 \leq \alpha_2, \alpha_3$ and $\alpha_2 + \alpha_3 = 1$, which clearly has only one feasible point: $\alpha_2 = \alpha_3 = 0.5$. The corresponding terms are $H'_1 = h_{1,2} = Z_1 X_2 - X_2$ and $H'_2 = h_{2,3} = -X_2 + X_2 Z_3$. Both of these terms are stoquastic and they satisfy $H = \sum_a H'_a$.

**Curing using Pauli operators.** In the next subsection, we show that conjugating a Hamiltonian by a tensor product of Pauli $X$ operators or identity operators only shuffles the off-diagonal elements without changing their values. Recalling that $Y = iXZ$, we thus conclude that choosing between Pauli operators to cure a Hamiltonian is equivalent to choosing between $I$ and $Z$ operators. Therefore, given local terms of a $k$-local Hamiltonian $\{H_a\}$ as input, the goal is to find a string $x = (x_1, \ldots, x_n)$ such that $U = \otimes_{i=1}^n Z^{x_i}$ cures each of the local terms $\{H_a\}$ separately.

Each multi-qubit Pauli operator in $H_a$ can be decomposed into $X$ components and $Z$ components. We group all the terms in each $H_a$ that share the same $X$ component. For example, if $H_a$ includes $Y_1 Y_2$, $3X_1 X_2$, and $X_1 X_2 Z_3$, we combine them into one single term $X_1 X_2(-Z_1 Z_2 + 3 + Z_3)$. Conjugating this term with $U$ yields $(-1)^{x_1 + x_2} X_1 X_2(-Z_1 Z_2 + 3 + Z_3)$. As terms with different $X$ components do not correspond to overlapping off-diagonal elements in $H_a$, the combined $Z$ part fixes a constraint on $\{x_i\}$ based on the positivity or negativity of all its elements (if the combined $Z$ part has both positive and negative elements, we conclude that there is no $U$ that can cure the input $H$). In this example, $(-1)^{x_1 + x_2} X_1 X_2(-Z_1 Z_2 + 3 + Z_3)$ becomes stoquastic iff $x_1 + x_2 \equiv 1 \mod 2$.

We combine all these linear equations in mod 2 that are generated from terms with different $X$ components, and solve for a satisfying $x$. This can be done efficiently, e.g., using Gaussian elimination. The absence of a consistent solution implies the absence of a curing Pauli group element.

As the dimension of each of the local terms $\{H_a\}$ is independent of the number of qubits $n$, and there are at most $poly(n)$ of these terms, the entire procedure takes $poly(n)$ time.

**Conjugation by a product of $X$ operators.** Here, we show that conjugating a Hamiltonian by a tensor product of $X$'s or identity operators only shuffles the off-diagonal elements without changing their values.

**Lemma 3** *Let $U_X = X^{a_1} \otimes \ldots \otimes X^{a_n}$, where $a_i \in \{0, 1\}$. The set of off-diagonal elements of a general $2^n \times 2^n$ matrix $B$, OFF$(B) = \{[B]_{ij} | i, j \in \{0,1\}^n, i \neq j\}$, is equal to the set of off-diagonal elements of $U_X B U_X$ for all possible $\{a_i\}$.*

*Proof.* Let $a = (a_1, \ldots, a_n)$. Similarly, let $i$ and $j$ represent $n$-bit strings. The elements of the matrix $U_X B U_X$ are

$$\begin{aligned}
\langle i | U_X B U_X | j \rangle &= \langle i_1, \ldots, i_n | U_X B U_X | j_1, \ldots, j_n \rangle \\
&= \langle i_1 \odot a_1, \ldots, i_n \odot a_1 | B | j_1 \odot a_1, \ldots, j_n \odot a_1 \rangle \\
&= \langle i \odot a | B | j \odot a \rangle
\end{aligned} \tag{32}$$

where $\odot$ denotes the XOR operation. Clearly for any fixed $a$, we have $i \neq j \Leftrightarrow i \odot a \neq j \odot a$ and therefore

$$\begin{aligned}
\text{OFF}(U_X B U_X) &= \{[B]_{i \odot a, j \odot a} | i, j \in \{0, 1\}^n, i \neq j\} \\
&= \{[B]_{i \odot a, j \odot a} | i \odot a, j \odot a \in \{0, 1\}^n, i \odot a \neq j \odot a\} \\
&= \{[B]_{i'j'} | i', j' \in \{0, 1\}^n, i' \neq j'\} \{[B]_{i'j'} | i', j' \in \{0, 1\}^n, i' \neq j'\} \\
&= \text{OFF}(B)
\end{aligned} \tag{33}$$

A similar argument proves that $U_X$ shuffles the diagonal elements: $\text{DIA}(U_X B U_X) = \text{DIA}(B)$.

Therefore, conjugating a Hamiltonian by $U_X$ does not change the set of inequalities one needs to solve to make a Hamiltonian stoquastic. As a consequence, whenever we want to pick $u_i$ as a solution, we can instead choose $X u_i$.

**One-local rotations are not enough**. Consider, e.g., the three Hamiltonians

$$H_{ij} = Z_i Z_j + X_i X_j, \quad (i, j) \in \{(1, 2), (2, 3), (3, 1)\}. \tag{34}$$

The sum of any pair of these Hamiltonians can be cured by single-qubit unitaries (e.g., $H_2 = H_{1,2} + H_{2,3}$ can be cured by applying $U = Z_2$).

In contrast, the frustrated Hamiltonian $H = H_{12} + H_{23} + H_{13}$ cannot be cured using any combination of single-qubit rotations. To see this, we first note that the partial trace of a stoquastic Hamiltonian is necessarily stoquastic. By partial trace over single qubits of $H$, we conclude that in order for $H$ to be stoquastic, all three $H_{ij}$'s must be stoquastic. To find all the solutions that convert each of these Hamiltonians into a stoquastic Hamiltonian, we expand $R_i(\theta_i) \otimes R_j(\theta_j) H_{ij} R_i^T(\theta_i) \otimes R_j^T(\theta_j)$ and note that it has $\pm \sin 2(\theta_i - \theta_j)$ and $\cos 2(\theta_i - \theta_j)$ as off-diagonal elements. Demanding that the rotated Hamiltonians are all stoquastic [so that $\sin 2(\theta_i - \theta_j) = 0$] forces $\cos 2(\theta_i - \theta_j) = -1 \ \forall (i, j) \in \{(1, 2), (2, 3), (3, 1)\}$. But this set of constraints does not have a feasible point. To see this note that

$$\begin{aligned}
\cos 2(\theta_1 - \theta_3) &= \cos 2(\theta_1 - \theta_2) \cos 2(\theta_1 - \theta_3) \\
&\quad - \sin 2(\theta_1 - \theta_2) \sin 2(\theta_1 - \theta_3) \\
&= -1 \times -1 + 0 \times 0 = +1.
\end{aligned} \tag{35}$$

Therefore, $H$ cannot be made stoquastic using $2 \times 2$ rotational matrices. Because of the relation between rotation and orthogonal matrices discussed above, we conclude that $H$ cannot be made stoquastic using $2 \times 2$ orthogonal matrices.

**Encryption based on secretly stoquastic Hamiltonians**. By generating 3SAT instances with planted solutions (see, e.g., refs. [17,18]) and transforming these to non-stoquastic Hamiltonians via the mapping prescribed by Theorems 1 (or 2), one would be able to generate 3-local (or 6-local) Hamiltonians that are stoquastic, but are computationally hard to transform into a stoquastic form.

This construction may have cryptographic implications. For example, imagine planting a secret $n$-bit message in the (unique by design) ground state of a stoquastic Hamiltonian. Since the solution is planted, Alice automatically knows it. She checks that QMC can find the ground state in a prescribed amount of time $\tau(n)$, and if this is not the case, she generates a new, random, stoquastic Hamiltonian with the same planted solution and checks again, etc., until this condition is met. Alice and Bob pre-share the secret key, i.e., the curing transformation, and after they separate, Alice transmits only the $O(n^2)$ coefficients of the non-stoquastic Hamiltonians (transformed via the mapping prescribed by Theorem 1) for every new message she wishes to send to Bob. To discover Alice's secret message, Bob runs QMC on the cured Hamiltonian. Since Alice verified that QMC can find the ground state in polynomial time, Bob will also find the ground state in polynomial time.

This scheme should be viewed as merely suggestive of a cryptographic protocol, since as it stands it contains several potential loopholes: (i) its security depends on the absence of efficient two-or-more qubit curing transformations, as well as the absence of algorithms other than QMC that can efficiently find the ground state of the non-stoquastic Hamiltonians generated by Alice; (ii) the fact that Alice must start from Hamiltonians for which the ground state can be found in polynomial time may make the curing problem easy as well; (iii) this scheme transmits an $n$-bit message using $r n^2$ message bits, where $r$ is the number of bits required to specify the $n^2$ coefficients of the transmitted non-stoquastic Hamiltonian, so it is less efficient than a one-time pad. Additional

research is needed to improve this into a scheme that overcomes these objections.

## Data availability

Data sharing was not applicable to this article, as no datasets were generated or analyzed during the current study.

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

## Acknowledgements

The research is based upon work (partially) supported by the Office of the Director of National Intelligence (ODNI), Intelligence Advanced Research Projects Activity (IARPA), via the U.S. Army Research Office contract W911NF-17-C-0050. The views and conclusions contained herein are those of the authors and should not be interpreted as necessarily representing the official policies or endorsements, either expressed or implied, of the ODNI, IARPA, or the U.S. Government. The U.S. Government is authorized to reproduce and distribute reprints for Governmental purposes notwithstanding any copyright annotation thereon. We thank Ehsan Emamjomeh-Zadeh, Iman Marvian, Evgeny Mozgunov, Ben Reichardt, and Federico Spedalieri for useful discussions.

## Author contributions

I.H. conceived of the project. M.M. devised most aspects of the technical proofs. I.H., D.A.L. and M.M. contributed equally to discussions and writing the paper.

## Additional information

**Competing interests:** The authors declare no competing interests.

