## [Peer Review File · Nature Communications]

Reviewers' comments:

Reviewer #1 (Remarks to the Author):

The paper considers quantum spin Hamiltonians with few-spin interactions and the problem of finding a local basis change that makes a given Hamiltonian "stoquastic". It is shown that this problem is NP-hard in the case when the basis change transformation is a tensor product of single-qubit Clifford gates or real unitary gates.

I find the problem studied in the paper very appealing. It arises naturally in the design of Quantum Monte Carlo algorithms where "stoquasticity" plays the major role and determines which Hamiltonians are easy and which are hard to simulate. Most of the previous work explored implications of stoquasticity for complexity of various simulation tasks. In contrast, the present paper explores the hardness of deciding whether a given Hamiltonian is stoquastic (modulo a local basis change).

The reduction to 3-SAT proposed by the authors is very clean and easy to understand. As far as I can check, the results are technically correct (although I was not able to verify all the steps in the proof of Lemma 2).

Speaking of weak sides, Hamiltonians constructed in the paper are somewhat contrived and do not resemble any physical model studied before. This may limit the interest of the new results for the physics community. It would be nice to comment whether the authors expect the NP-hardness result to hold for more physical Hamiltonians, e.g. translation-invariant and/or 2-local Hamiltonians. Secondly, I feel that the authors should give more credit to Hastings's 2016 paper (reference 16). Hastings was the first who posed the question of whether the sign problem can be cured by a low-depth quantum circuit, which is the main question studied in the present paper. I have several other comments of more technical nature that the authors should address (see below).

My overall impression is that the paper initiates an important research direction and reports a highly non-trivial and interesting observation. I expect it to be of great interest for researchers working on computational physics and complexity theory. Therefore I recommend acceptance.

Technical comments:

Introduction, page 1: the claim that the sign problem is "the single most important unresolved challenge in quantum many-body simulations" sounds too strong. I would suggest to tone this down.

Proof of Lemma 1, part (ii): the authors implicitly assume that a 3-SAT formula contains at most one clause for each triple of bits. Is it known that 3-SAT remains NP-complete under this restriction? This should be clarified.

Proof of Lemma 1, paragraph below Eq. 6: it might be worth elaborating on how large the coefficient c should be. The authors choose $c=O(1) M$ but it is not clear to me why this is "large enough" because $H_{\{3SAT\}}$ may contain $O(M^2)$ clauses acting on any given qubit.

A related question: if c is sufficiently large then the ground energy of the full Hamiltonian can be efficiently computed using the perturbation theory (by treating $H_{\{3SAT\}}$ as a perturbation of cH_0). If this is the case, nobody would care whether the full Hamiltonian is stoquastic or non-stoquastic. It would be nice to comment whether there is a regime when the sign curing problem is NP-hard but the ground energy cannot be easily computed by perturbation theory.

Page 7, Lemma 2: the notations are very confusing. The tensor product in the definition of H' is different from the tensor product in the definition of a curing rotation. Also, I believe one has to add a condition that the Hamiltonians H_z, H_x etc are non-zero (otherwise the lemma is trivially false).

Page 1, Definition 1: I feel that introduction is not the best place for formal definitions.

Besides, the definition does not explain what are the terms H_a in the sum.

Perhaps one can introduce stoquastic Hamiltonians in a less formal way.

Page 2, Definition 2: it appears that one important condition is missing here. Namely, the classical algorithm must be able to compute the transformed Hamiltonian H' efficiently. Since the definition does not assume anything about the unitary transformation U (like a constant-depth), it may be hard to compute H' , even if it is promised to be local.

Theorem 1: I am curious if the authors expect the theorem to hold for 2-local Hamiltonians as well?

This is arguably the most important case from the physics perspective. It might be worth adding a comment along these lines.

Lemma 1: I don't think W_i needs a subscript here (since u_i was defined as a single-qubit operator).

Page 3, sentence above Eq. 5: "as" should be "is"

Reviewer #2 (Remarks to the Author):

This is a long awaited, essential paper classifying the complexity of a variant of the sign problem that plagues essential numerical methods for quantum systems - Quantum Monte Carlo (QMC). Previous work [7, PRL] attempting to formalize common knowledge that dealing with the sign problem is hard was simply handwavy and without proper statements - it only underlined that the problem is hard. Here, the authors finally give a proper complexity theoretic theorem about this. I only wish I proved this first (-; Others have immediately continued this line of research [15].

The sign problem (varying signs of off-diagonal elements in a Hamiltonian, resulting in varying sign of the amplitudes of a ground states, not interpretable directly as probabilities, which means one can not directly apply QMC) can be dealt with by clever tricks, but thanks to instabilities due to the varying sign that appears in the denominator of some expressions, they require many sampling steps and result in uncertain precision. This is why any other way of dealing with or understanding the sign problem is a very desirable thing.

The paper's main claim is: Figuring out if one can remove the sign problem for certain (3-local) Hamiltonians by single qubit Clifford or OG matrices is NP complete. Thus, we should not expect effective algorithms for this. Moreover, this is just a statement about the first step - removing the sign problem, but not about the complexity of the stoquastic problem that follows, and still needs to be dealt with by QMC. This distinction is important, as [7] was muddling it up together with the hardness of sign problem removal.

As such, it opens the way to investigation of hardness, but also systematic ways of curing the sign problem, which makes it exciting for a large group of researchers. It also contributes to the much

needed dialogue between condensed matter theory and complexity theory. Therefore, I gladly recommend it for publication in Nature Comms.

The technical side of the paper is solid, as far as I could check, and the paper is well written. The authors show NP-hardness by starting with an instance of the NP-complete problem 3SAT, and preparing a 3-local Hamiltonian in such a way that curing its nonstoquasticity locally would mean a choice of a bit string that solves the 3SAT instance. Inclusion in NP is straightforward - just take the list of local transformations and check if the local Hamiltonian terms become stoquastic. Altogether, this implies NP-completeness. Clean and clear.

Reviewer #3 (Remarks to the Author):

In this manuscript, the authors consider the problem of determining whether there exists a unitary (from a particular simple class, the single-qubit Clifford group) that rotates a given matrix into a so-called stoquastic matrix. The authors prove that, even if we only look to decide whether there exists a unitary in this class that solves the problem, it is still NP-hard.

This is an interesting paper and certainly worthy of publication, but I am not sure that the novelty or the methods used merit its publication in this particular journal. That is, I believe that while this may be an interesting result for the quantum information community, it is already well-established (and may be well-known) to complexity theorists. Nonetheless, that the presentation follows the expectations and derives the result rather easily is a strength of the paper.

In particular, I worry that the off-diagonal terms in the Hamiltonians under consideration in this manuscript often correspond to adjacency matrices of signed graphs, including the example problem the authors lay out initially. It is well-known that if one begins with a signed adjacency matrix (terms being all +/-1), the algorithm for finding the "curing" unitary requires walking along every vertex in the graph. (This is the problem of detecting balance in a signed graph.) In the quantum setting, because the number of vertices becomes exponentially large, it is not at all surprising that finding the unitary even in this limited case is exceptionally difficult. It would require visiting 2^n sites, where n is the number of qubits. That is, insert any frustrated triangle at the boundary of any balanced graph and the graph becomes unbalanced. In the worst case, we now need to visit all 2^n

sites before determining that the graph is unbalanced. This also seems to be a more general claim than that in the paper, since this classical method would be restricted to just the Identity and Z operators. It is unclear why someone would expect that the locality of the Hamiltonian would make the problem easier, as each vertex still must be visited at least once.

The authors should see, e.g. M. Yannakakis, Edge-deletion problems, *SIAM Journal on Computing*, 10 (1981), pp. 297–309., arXiv:1611.09030, Frank Harary and Jerald A Kabell. “A simple algorithm to detect balance in signed graphs”. In: *Mathematical Social*

Sciences 1.1 (1980), pp. 131–136. issn: 0165-4896. There are many other papers exploring this topic.

Again, I think this paper is quite worthy of publication and will genuinely be of interest, however I do not believe the result will be terribly surprising beyond the quantum community.

Reviewer 1 (Remarks to the Author):

The paper considers quantum spin Hamiltonians with few-spin interactions and the problem of finding a local basis change that makes a given Hamiltonian "stoquastic". It is shown that this problem is NP-hard in the case when the basis change transformation is a tensor product of single-qubit Clifford gates or real unitary gates.

I find the problem studied in the paper very appealing. It arises naturally in the design of Quantum Monte Carlo algorithms where "stoquasticity" plays the major role and determines which Hamiltonians are easy and which are hard to simulate. Most of the previous work explored implications of stoquasticity for complexity of various simulation tasks. In contrast, the present paper explores the hardness of deciding whether a given Hamiltonian is stoquastic (modulo a local basis change). The reduction to 3-SAT proposed by the authors is very clean and easy to understand. As far as I can check, the results are technically correct (although I was not able to verify all the steps in the proof of Lemma 2).

Speaking of weak sides, Hamiltonians constructed in the paper are somewhat contrived and do not resemble any physical model studied before. This may limit the interest of the new results for the physics community. It would be nice to comment whether the authors expect the NP-hardness result to hold for more physical Hamiltonians, e.g. translation-invariant and/or 2-local Hamiltonians.

We agree that it would be very interesting to apply and extend our study to simpler Hamiltonians, such as 2-local Hamiltonians. We are confident that the research program initiated here, and even the mapping we introduced, can be generalized to more experimentally relevant Hamiltonians. More specifically, for the case of 2-local Hamiltonians, we will provide evidence for this under the "technical comments", where the reviewer brought up the same point. But before that, two points are in order.

As is common for complexity-theoretic results, it is more insightful to first prove the hardness using more intuitive constructions and then, with extra effort, try to extend the result to the family of problems that are of particular interest for some given setup. A relevant example is the QMA-hardness of estimating the ground state energy of local Hamiltonians. The original proof for 5-local Hamiltonians is more straightforward and illuminating, but extending it to 2-local Hamiltonians took a lot of effort. In that respect, our constructed Hamiltonians are real, 3-local (and 6-local) Hamiltonians consisting of only Z and X Pauli operators. Fortunately, as we describe below, we believe that the mapping we introduced here is powerful and has the potential to be used to prove the hardness of other simpler Hamiltonians.

Let us emphasize that our direct mapping to 3-SAT is also powerful in another sense: any restricted family of 3-SAT that is still NP-complete directly provides us with a restriction on the family of Hamiltonians for which the curing problem is still NP-complete. Such examples include Planar 3SAT and a variant of 3SAT wherein each variable is restricted to appear at most four times.¹

Secondly, I feel that the authors should give more credit to Hastings's 2016 paper (reference 16). Hastings was the first who posed the question of whether the sign problem can be cured by a low-depth quantum circuit, which is the main question studied in the present paper.

We added a reference to this paper at the point where we motivate the rotations with constant depth circuits.

To be clear, in the introduction, we separated two different scenarios for the curing problems. One problem asks about the computational complexity of curing, and motivated by that we defined efficiently curable Hamiltonians. The other problem is the important question raised by Hastings's 2016 paper, whether it is possible to have Hamiltonians that are not curable with some reasonable size circuit. Note that the existence of such a Hamiltonians does not tell us anything about the computational complexity of finding curing transformation. This is evident from the $H = X_1 Z_2$ example we provided in the main text.

I have several other comments of more technical nature that the authors should address (see below).

My overall impression is that the paper initiates an important research direction and reports a highly non-trivial and interesting observation. I expect it to be of great interest for researchers working on computational physics and complexity theory. Therefore I recommend acceptance.

We are grateful for the positive feedback and for recommending acceptance.

Technical comments:

- Introduction, page 1: the claim that the sign problem is "the single most important unresolved challenge in quantum many-body simulations" sounds too strong. I would suggest to tone this down.

We agree and modified the sentence.

- Proof of Lemma 1, part (ii): the authors implicitly assume that a 3-SAT formula contains at

¹ For example, see <http://courses.csail.mit.edu/6.890/fall14/scribe/lec7.pdf>, https://www.csie.ntu.edu.tw/~lyuu/complexity/2004/c_20041027.pdf, <http://courses.csail.mit.edu/6.890/fall14/scribe/lec4.pdf>.

most one clause for each triple of bits. Is it known that 3-SAT remains NP-complete under this restriction ? This should be clarified.

Without loss of generality, we assume there are no repeated clauses. In that case, a choice of variables that makes a clause False, definitely makes another clause on the same variables True. A satisfied Clause does not generate an XXX term. Therefore other clauses on the same variables cannot cancel the XXX resulted from the original unsatisfied clause. We added an explanation to clarify this. In this argument, we have implicitly assumed that every clause has three variables. We updated the text spelling this out, with a reference to show that this slightly modified version of 3SAT, called Exact 3-SAT,² is still NP-complete.

- Proof of Lemma 1, paragraph below Eq. 6: it might be worth elaborating on how large the coefficient c should be. The authors choose $c = O(1)M$ but it is not clear to me why this is “large enough” because H_{3SAT} may contain $O(M^2)$ clauses acting on any given qubit.

A related question: if c is sufficiently large then the ground energy of the full Hamiltonian can be efficiently computed using the perturbation theory (by treating H_{3SAT} as a perturbation of cH_0). If this is the case, nobody would care whether the full Hamiltonian is stoquastic or non-stoquastic. It would be nice to comment whether there is a regime when the sign curing problem is NP-hard but the ground energy cannot be easily computed by perturbation theory.

Below Eq.(4), we defined M to be the total number of clauses. Therefore, each variable (qubit) can be acted on by at most M clauses. Each clause-Hamiltonian contains some constant number of Pauli operators, therefore $c = O(1)M$ suffices, as we have clarified in the new version.

We agree that this might be overkill; we provided this as a simple upper bound. We only need to choose c for each variable to be the number of clauses it participates in. To reduce this number, we can make the simple observation that even a restricted variant of 3SAT with each variable restricted to appear at most in 4 clauses, is still NP-Complete.³ Therefore, for these problems we can take c to be a constant.

Also, for the second theorem, we already provided a proof for Hamiltonians with constant couplings (see below Eq.10).

² T.J. Cook 71

³ See Tovey, C.A., 1984. “A simplified NP-complete satisfiability problem.” *Discrete applied mathematics*, 8(1), pp.85-89.

- Page 7, Lemma 2: the notations are very confusing. The tensor product in the definition of H' is different from the tensor product in the definition of a curing rotation.

Also, I believe one has to add a condition that the Hamiltonians H_z, H_x etc are non-zero (otherwise the lemma is trivially false).

We agree, and rewrote this entire section in order to clarify Lemma 2. We hope that it is now easier to follow.

By moving the two conditions which were presented at the beginning of page 8 into the statement of Lemma 2, we hope we have clarified what was meant by “ c is an appropriately chosen constant.” Also, rather than choosing $c = 1$ at the very end, we removed any discussion about c , and just showed the result with fixing $c = 1$ from the beginning. The technical results are unchanged after these modifications.

- Page 1, Definition 1: I feel that introduction is not the best place for formal definitions. Besides, the definition does not explain what are the terms H_a in the sum. Perhaps one can introduce stoquastic Hamiltonians in a less formal way.

The definition of stoquastic Hamiltonians is a central motivation for our work. Quoting the widely used formal definition of stoquasticity in the Introduction helps us introduce the benefit of our definition, which considers the computational cost of curing. This then allows us to present our hardness results.

- Page 2, Definition 2: it appears that one important condition is missing here. Namely, the classical algorithm must be able to compute the transformed Hamiltonian H' efficiently. Since the definition does not assume anything about the unitary transformation U (like a constant-depth), it may be hard to compute H' , even if it is promised to be local.

We agree and modified the sentence to demand that the algorithm also output a description of H' efficiently:

“the algorithm can find a unitary U and a Hamiltonians H' with the property...”

- Theorem 1: I am curious if the authors expect the theorem to hold for 2-local Hamiltonians as well ? This is arguably the most important case from the physics perspective. It might be worth adding a comment along these lines.

Yes, we do expect this. The key observation in this direction is that our mapping, between the choice of Hadamard or Identity and a 3SAT clause, can be constructed even using 2-local

Hamiltonians. For example, consider the following 2-local Hamiltonian:

$$H_{ijk}^{(111)} = -2(X_i + X_j + X_k) - (Z_i + Z_j + Z_k) - (X_1 Z_2 + X_1 Z_3 + X_2 Z_1 + X_2 Z_3 + X_3 Z_2 + X_3 Z_1).$$

Going through the whole set of possible choices of applying the Hadamard or not, one can confirm that:

$$(W_i^{x_i} W_j^{x_j} W_j^{x_j}) H_{ijk}^{(111)} (W_j^{x_j} W_j^{x_j} W_j^{x_j})$$

is stoquastic for any choice of binary variables x except for $(x_i, x_j, x_k) = (1, 1, 1)$. Therefore, this 2-local Hamiltonian would correspond to the 3-SAT clause $C_{ijk}^{(111)}$. Other possible clauses can be constructed similarly.

To complete the proof, i.e., to show that the hardness result does not go away when we add up the clause-Hamiltonian terms, would require a different approach than the one presented in our paper. Therefore we do not claim that this is a proof for the NP-completeness of 2-local Hamiltonians. But, this gives a strong indication that our approach can work beyond 3-local Hamiltonians. Establishing such a proof is among the many open problems that are initiated by this work.

- Lemma 1: I don't think W_j needs a subscript here (since u_i was defined as a single-qubit operator).
- Page 3, sentence above Eq. 5: "as" should be "is"

We agree with both comments and updated the text accordingly.

We sincerely thank the Reviewer for carefully reading our manuscript and the many very constructive and useful comments.

Reviewer 2 (Remarks to the Author):

This is a long awaited, essential paper classifying the complexity of a variant of the sign problem that plagues essential numerical methods for quantum systems - Quantum Monte Carlo (QMC). Previous work [7, PRL] attempting to formalize common knowledge that dealing with the sign problem is hard was simply handwavy and without proper statements - it only underlined that the problem is hard. Here, the authors finally give a proper complexity theoretic theorem about this. I only wish I proved this first (-; Others have immediately continued this line of research [15].

The sign problem (varying signs of off-diagonal elements in a Hamiltonian, resulting in varying sign of the amplitudes of a ground states, not interpretable directly as probabilities, which means one can not directly apply QMC) can be dealt with by clever tricks, but thanks to instabilities due to the varying sign that appears in the denominator of some expressions, they require many sampling steps and result in uncertain precision. This is why any other way of dealing with or understanding the sign problem is a very desirable thing.

The paper's main claim is: Figuring out if one can remove the sign problem for certain (3-local) Hamiltonians by single qubit Clifford or OG matrices is NP complete. Thus, we should not expect effective algorithms for this. Moreover, this is just a statement about the first step - removing the sign problem, but not about the complexity of the stoquastic problem that follows, and still needs to be dealt with by QMC. This distinction is important, as [7] was muddling it up together with the hardness of sign problem removal.

As such, it opens the way to investigation of hardness, but also systematic ways of curing the sign problem, which makes it exciting for a large group of researchers. It also contributes to the much needed dialogue between condensed matter theory and complexity theory. Therefore, I gladly recommend it for publication in Nature Comms.

The technical side of the paper is solid, as far as I could check, and the paper is well written. The authors show NP-hardness by starting with an instance of the NP-complete problem 3SAT, and preparing a 3-local Hamiltonian in such a way that curing its nonstoquasticity locally would mean a choice of a bit string that solves the 3SAT instance. Inclusion in NP is straightforward - just take the list of local transformations and check if the local Hamiltonian terms become stoquastic. Altogether, this implies NP-completeness. Clean and clear.

We sincerely appreciate the positive and encouraging feedback of the Reviewer.

Reviewer 3 (Remarks to the Author):

In this manuscript, the authors consider the problem of determining whether there exists a unitary (from a particular simple class, the single-qubit Clifford group) that rotates a given matrix into a so-called stoquastic matrix. The authors prove that, even if we only look to decide whether there exists a unitary in this class that solves the problem, it is still NP-hard.

This is an interesting paper and certainly worthy of publication, but I am not sure that the novelty or the methods used merit its publication in this particular journal. That is, I believe that while this may be an interesting result for the quantum information community, it is already well-established (and may be well-known) to complexity theorists. Nonetheless, that the presentation follows the expectations and derives the result rather easily is a strength of the paper.

In particular, I worry that the off-diagonal terms in the Hamiltonians under consideration in this manuscript often correspond to adjacency matrices of signed graphs, including the example problem the authors lay out initially. It is well-known that if one begins with a signed adjacency matrix (terms being all ± 1), the algorithm for finding the "curing" unitary requires walking along every vertex in the graph. (This is the problem of detecting balance in a signed graph.) In the quantum setting, because the number of vertices becomes exponentially large, it is not at all surprising that finding the unitary even in this limited case is exceptionally difficult. It would require visiting 2^n sites, where n is the number of qubits. That is, insert any frustrated triangle at the boundary of any balanced graph and the graph becomes unbalanced. In the worst case, we now need to visit all 2^n sites before determining that the graph is unbalanced.

This also seems to be a more general claim than that in the paper, since this classical method would be restricted to just the Identity and Z operators. It is unclear why someone would expect that the locality of the Hamiltonian would make the problem easier, as each vertex still must be visited at least once.

The authors should see, e.g. M. Yannakakis, Edge-deletion problems, SIAM Journal on Computing, 10 (1981), pp. 297309., arXiv:1611.09030, Frank Harary and Jerald A Kabell. A simple algorithm to detect balance in signed graphs. In: Mathematical Social Sciences 1.1 (1980), pp. 131136. issn: 0165-4896. There are many other papers exploring this topic.

Again, I think this paper is quite worthy of publication and will genuinely be of interest, however I do not believe the result will be terribly surprising beyond the quantum community.

The Reviewer makes a thought provoking observation, but the problem we consider here, curing a local Hamiltonian using local rotation, is fundamentally different from the problem of balancing a signed graph. A good way to appreciate the difference is to focus on the problem of choosing

between Identity and Z proposed by the Reviewer. In Appendix B of our manuscript, we already considered this very problem (even more generally, we considered choosing between Pauli operators) and already showed there that this problem can be solved *efficiently*, contrary to the Reviewer's expectation that it is exponentially hard.

As we extensively discussed in the introduction, the relevant problem for Quantum Monte Carlo simulations and also for the definition of the complexity class StoqMA is whether we can find a set of local Hamiltonians $\{H_a\}$, whose off-diagonal elements are non-positive. Each $\{H_a\}$ has a constant size and there are only polynomially many of these terms. In contrast to the sign balancing problem, here we don't have to fix an exponential number of off-diagonal elements, where each essentially has to be fixed individually.

In addition, the action of physically relevant transformations (that keep the partition function unchanged) would alter several off-diagonal elements at once. The balancing of signed graphs may therefore be thought of as a very restrictive form of curing, namely, one in which one is *not* allowed to rotate the basis but only phases. However, as we show, the sign problem may be cured by the more general class of transformations, namely, the rotation of basis.

Hopefully it is clear from these observations that the problem we studied here is unrelated to the problem of balancing a signed graph.

REVIEWERS' COMMENTS:

Reviewer #1 (Remarks to the Author):

The authors have addressed most of the comments from my review.

There is still one unresolved issue: operators H_a in Definition 1 are undefined. I believe the authors meant that H_a are operators acting non-trivially only on a few qubits. This should be explicitly stated.

Apart from that, I find the new version ready for publication.

Reviewer #3 (Remarks to the Author):

I believe the authors have successfully addressed my main concern with this paper, though I had to look at the arxiv version (not this version) to find the result that does so. Appendix B does not exist in the file I was provided.

That said, I still do not believe that the authors appreciate the sign-balancing problem (and therefore some of the implications of what is a fairly simple argument of their own). Specifically, the sign-balancing problem is basically concerned with determining the ground-state energy of a Hamiltonian, in a very similar fashion to StoqMA, however under different constraints. The arxiv version shows that efficient curing is indeed possible in these simple cases.

Although their proof is rather simple, I actually find that result a bit surprising and interesting in itself, since it effectively provides an algorithm for balancing graphs provided that they correspond to Cayley graphs with small-ish groups. In other words, this effectively defines and shows how to balance a large class of graphs, provided they have some (known) compact description.

This is not yet known in the corresponding literature, though I would still have liked to see the point made clearly in this manuscript. Nonetheless, the authors do successfully address my primary concern. (I do not know if that appendix will appear in this version of the paper, however I think it should.)

Below please find our detailed reply to the reports, with the Reviewers' comments in black and our responses in blue.

We greatly appreciate the Reviewers' feedback, which has helped to improve our presentation.

Reviewer 1 (Remarks to the Author):

The authors have addressed most of the comments from my review. There is still one unresolved issue: operators H_a in Definition 1 are undefined. I believe the authors meant that H_a are operators acting non-trivially only on a few qubits. This should be explicitly stated.

Apart from that, I find the new version ready for publication.

We modified the Definition 1 to reflect this: “all the local terms H_a ” was added.

Reviewer 3 (Remarks to the Author):

I believe the authors have successfully addressed my main concern with this paper, though I had to look at the arxiv version (not this version) to find the result that does so. Appendix B does not exist in the file I was provided.

That said, I still do not believe that the authors appreciate the sign-balancing problem (and therefore some of the implications of what is a fairly simple argument of their own). Specifically, the sign-balancing problem is basically concerned with determining the ground-state energy of a Hamiltonian, in a very similar fashion to StoqMA, however under different constraints. The arxiv version shows that efficient curing is indeed possible in these simple cases.

Although their proof is rather simple, I actually find that result a bit surprising and interesting in itself, since it effectively provides an algorithm for balancing graphs provided that they correspond to Cayley graphs with small-ish groups. In other words, this effectively defines and shows how to balance a large class of graphs, provided they have some (known) compact description.

This is not yet known in the corresponding literature, though I would still have liked to see the point made clearly in this manuscript. Nonetheless, the authors do successfully address my primary concern. (I do not know if that appendix will appear in this version of the paper, however I think it should.)

The appendix is now included in the main text.